# Evolution of Anti-RBD IgG Avidity following SARS-CoV-2 Infection

**DOI:** 10.3390/v14030532

**Published:** 2022-03-04

**Authors:** Alexandra Tauzin, Gabrielle Gendron-Lepage, Manon Nayrac, Sai Priya Anand, Catherine Bourassa, Halima Medjahed, Guillaume Goyette, Mathieu Dubé, Renée Bazin, Daniel E. Kaufmann, Andrés Finzi

**Affiliations:** 1Centre de Recherche du CHUM, Montreal, QC H2X 0A9, Canada; alexandra_tauzin@hotmail.fr (A.T.); gabrielle.gendron-lepage.chum@ssss.gouv.qc.ca (G.G.-L.); manonnayrac@gmail.com (M.N.); sai.anand@mail.mcgill.ca (S.P.A.); catherine.bourassa.chum@ssss.gouv.qc.ca (C.B.); halima.medjahed.chum@ssss.gouv.qc.ca (H.M.); guillaume@biotechconnect.ca (G.G.); mathieudubephd@gmail.com (M.D.); daniel.kaufmann@umontreal.ca (D.E.K.); 2Département de Microbiologie, Infectiologie et Immunologie, Université de Montréal, Montreal, QC H3C 3J7, Canada; 3Department of Microbiology and Immunology, McGill University, Montreal, QC H3A 0G4, Canada; 4Héma-Québec, Affaires Médicales et Innovation, Montreal, QC G1V5C3, Canada; renee.bazin@hema-quebec.qc.ca; 5Département de Médecine, Université de Montréal, Montreal, QC H3C 3J7, Canada

**Keywords:** COVID-19, SARS-CoV-2, convalescent plasma, antibodies, avidity, receptor-binding domain

## Abstract

SARS-CoV-2 infection rapidly elicits anti-Spike antibodies whose quantity in plasma gradually declines upon resolution of symptoms. This decline is part of the evolution of an immune response leading to B cell differentiation into short-lived antibody-secreting cells or resting memory B cells. At the same time, the ongoing class switch and antibody maturation processes occurring in germinal centers lead to the selection of B cell clones secreting antibodies with higher affinity for their cognate antigen, thereby improving their functional activity. To determine whether the decline in SARS-CoV-2 antibodies is paralleled with an increase in avidity of the anti-viral antibodies produced, we developed a simple assay to measure the avidity of anti-receptor binding domain (RBD) IgG elicited by SARS-CoV-2 infection. We longitudinally followed a cohort of 29 convalescent donors with blood samples collected between 6- and 32-weeks post-symptoms onset. We observed that, while the level of antibodies declines over time, the anti-RBD avidity progressively increases and correlates with the B cell class switch. Additionally, we observed that anti-RBD avidity increased similarly after SARS-CoV-2 mRNA vaccination and after SARS-CoV-2 infection. Our results suggest that anti-RBD IgG avidity determination could be a surrogate assay for antibody affinity maturation and, thus, suitable for studying humoral responses elicited by natural infection and/or vaccination.

## 1. Introduction

Infection with the severe acute respiratory syndrome coronavirus 2 (SARS-CoV-2), responsible for the coronavirus disease 2019 (COVID-19), leads to the rapid induction of antibodies (Abs) against this virus. The main target of the antibody response is the Spike (S) glycoprotein which mediates viral entry and is exposed at the surface of viral particles and infected cells [1,2,3]. Its receptor-binding domain (RBD) is responsible for the interaction with the human angiotensin-converting enzyme 2 (ACE2) receptor to initiate the fusion process [3,4,5]. The RBD also represents the main target of vaccine and infection-elicited neutralizing Abs [6,7]. We and other researchers have observed that the level of overall anti-S and anti-RBD Abs gradually decline, starting a few weeks following post symptoms onset (PSO) or a few weeks after vaccination [8,9,10,11,12,13,14,15]. Several studies have shown the generation of antibodies with higher affinity for RBD during the same period, indicative of germinal center formation and ongoing antibody maturation [16,17,18,19]. Antibody-mediated antiviral activity in plasma is a very complex process and depends on several factors, including the overall levels of antibodies, their respective affinity and avidity for their cognate antigen, and the functional interplay among them [20]. It is well established that after infection or vaccination, antibody avidity increases over time. This is due to B cell maturation that occurs in the germinal center (GC) leading to a maturation of Abs and enhancing their affinity (binding strength) for their cognate antigen [21]. While studying the maturation of B cells and associated Abs requires complex technical methods, we developed a simple and robust assay to measure the avidity (overall binding strength) of a polyclonal population of anti-RBD IgG. In this study, we used this assay to longitudinally measure the evolution of the anti-RBD avidity on a cohort of COVID-19 convalescent donors. We also used the same assay to measure the increase in avidity of antibodies elicited after one dose of an mRNA vaccine over a period of 12 weeks.

## 2. Materials and Methods

### 2.1. Ethics Statement

The study was conducted in accordance with the Declaration of Helsinki in terms of informed consent and approval by an appropriate institutional board. Peripheral blood mononuclear cells (PBMCs) and plasmas from convalescent and vaccinated individuals were obtained from donors who consented to participate in this research project at CHUM. The protocol was approved by the Ethics Committee of CHUM (protocol #19.381, approved on 25 March 2020). Donors met all eligibility criteria: previously confirmed COVID-19 infection and a complete resolution of symptoms for at least 14 days.

### 2.2. Plasma Samples, Primary Cells, and Antibodies

Plasmas and PBMCs were isolated from whole blood using centrifugation with a Ficoll gradient and stored at −80 °C and in liquid nitrogen until use. Plasmas were heat-inactivated for 1 h at 56 °C and stored at −80 °C until use in subsequent experiments. Healthy donors’ plasma, collected before the pandemic, was used as a negative control in ELISA assays (data not shown). The RBD-specific monoclonal antibody CR3022 was used as a positive control in our ELISA assays [8,9,10]. Horseradish peroxidase (HRP)-conjugated Abs (Invitrogen, Waltham, MA, USA) able to detect the Fc region of human IgG was used as secondary Abs to detect Ab binding in ELISA experiments.

### 2.3. Plasmids

The plasmid encoding the human coronavirus Spike of SARS-CoV-2 was kindly provided by Stefan Pöhlman and was previously reported [12].

### 2.4. Cell Line, Proteins Expression and Purification

FreeStyle 293 F cells (Invitrogen) were grown in FreeStyle 293F medium (Invitrogen) to a density of 10^6^ cells/mL at 37 °C with 8% CO_2_ under regular agitation (150 rpm). Cells were transfected with the plasmid coding for SARS-CoV-2 S RBD WT using an ExpiFectamine 293 transfection reagent, as directed by the manufacturer (Invitrogen). One week later, cells were pelleted and discarded. Supernatants were filtered using a 0.22 μm filter (Thermo Fisher Scientific, Waltham, MA, USA). The recombinant RBD proteins were purified using nickel affinity columns, as directed by the manufacturer (Invitrogen). The RBD preparations were dialyzed against phosphate-buffered saline (PBS) and stored in aliquots at −80 °C until further use. To assess purity, recombinant proteins were loaded on SDS-PAGE gels and stained with Coomassie Blue.

### 2.5. ELISA Assays

SARS-CoV-2 S RBD proteins (2.5 μg/mL) were prepared in PBS and adsorbed to plates (MaxiSorp Nunc, Thermo Fisher Scientific, Waltham, MA, USA.) overnight at 4 °C. Coated wells were subsequently blocked with a blocking buffer (Tris-buffered saline [TBS] containing 0.1% Tween20 and 2% BSA) for 1 h at room-temperature (RT). Wells were then washed four times with a washing buffer (Tris-buffered saline [TBS] containing 0.1% Tween20). CR3022 mAb (50 ng/mL) or a 1/250 dilution of plasma was prepared in a diluted solution of blocking buffer (0.1% BSA) and incubated in the RBD-coated wells for 90 min at RT. Plates were washed four times with a washing buffer followed by incubation with secondary Abs (diluted in a diluted solution of blocking buffer (0.4% BSA)) for 1 h at RT, followed by four washes. To calculate the RBD-avidity index, we performed a stringent ELISA, where the plates were washed at all steps with a chaotropic agent, 8M of urea, added to the washing buffer [12]. HRP enzyme activity was determined after the addition of a 1:1 mix of Western Lightning oxidizing and luminol reagents (Perkin Elmer Life Sciences, Waltham, MA, USA). Light emission was measured with a LB942 TriStar luminometer (Berthold Technologies, Bad Wildbad, Germany). The signal obtained was normalized using the signal obtained with CR3022 Ab in absence of urea present in each plate. The seropositivity threshold was established using the following formula: mean of pre-pandemic SARS-CoV-2 negative plasma+ (3 standard deviation of the mean of pre-pandemic SARS-CoV-2 negative plasma).

### 2.6. Detection of Antigen-Specific B Cells

To detect SARS-CoV-2-specific B cells, we conjugated recombinant RBD proteins with Alexa Fluor 488 or Alexa Fluor 594 (Thermo Fisher Scientific) according to the manufacturer’s protocol. Approximately 10^7^ frozen PBMCs from 12 convalescent donors were prepared at a final concentration of 14 × 10^6^ cells/mL in RPMI 1640 medium (GIBCO) supplemented with 10% of fetal bovine serum (VWR, Radnor, PA, USA), Penicillin-Streptomycin (GIBCO) and HEPES (GIBCO). After a rest of 2 h at 37 °C and 5% CO_2_, cells were stained using Aquavivid viability marker (GIBCO) in DPBS (GIBCO) at 4 °C for 20 min. The detection of SARS-CoV-2-antigen specific B cells was accomplished by adding the RBD probes to the following antibody cocktail: IgM BUV737, IgG BV421, CD3 BV480, CD56 BV480, CD14 BV480, CD16 BV480, and CD20 BV711, all from BD Biosciences, Franklin Lakes, NJ, USA and CD19 BV650 from Biolegend, San Diego, CA, USA. Staining was performed at 4 °C for 30 min and cells were fixed using 2% paraformaldehyde at 4 °C for 15 min. Stained PBMC samples were acquired with a Symphony cytometer (BD Biosciences) and analyzed using FlowJo v10.7.1 (TreeStar, Woodburn, OR, USA). The gating strategy is shown in Appendix A.

## 3. Results

### 3.1. Cohort of COVID-19 Convalescent Donors

We longitudinally followed the avidity of anti-RBD antibodies elicited by SARS-CoV-2 infection in a cohort of 29 COVID-19 convalescent donors during the first COVID-19 wave in the province of Quebec, Canada (Spring 2020). These donors were tested and determined to be SARS-CoV-2 positive using a reverse transcription PCR (RT-PCR) on nasopharyngeal swab specimens; they had mild to moderate disease symptoms and were not hospitalized. Convalescent participants were enrolled following two negative RT-PCR tests and a complete resolution of symptoms for at least 14 days before blood sampling. Blood samples were collected at four different time points PSO: 6 weeks, 11 weeks, 21 weeks, and 32 weeks. Basic demographic characteristics of the cohort are summarized in Table 1 and Figure 1.

### 3.2. RBD Avidity Assay

To measure the avidity of SARS-CoV-2 induced anti-RBD IgG, we adapted a previously described ELISA assay [12,22,23,24] (Figure 1). Briefly, plasma samples were tested by ELISA using two different washing conditions in parallel. In one condition, the washing buffer was supplemented with a chaotropic agent (urea 8M) but not in the other. The ELISA assay performed without stringent washes allowed us to measure the level of total SARS-CoV-2 RBD specific IgG. By adding urea to the washing buffer, only antibodies with a high affinity for the RBD remained bound (Figure 1 and Appendix A). These values were used to calculate an RBD avidity index using the following formula:(1)$$RBD\;avidity\;index=\frac{Level\;of\;anti-RBD\;IgG\;measured\;with\;urea}{Level\;of\;anti-RBD\;IgG\;measured\;without\;urea}\times 100$$

### 3.3. Evolution of Anti-RBD Avidity after Resolution of Symptoms

All convalescent individuals had detectable levels of anti-RBD IgG six weeks PSO, which gradually decreased overtime (Figure 2A and Appendix A) [8,9,10,25]. In contrast, anti-RBD avidity followed an inverse trajectory. It was relatively low with six weeks PSO but significantly increased thereafter in all donors (Figure 2B). The fold increase in RBD avidity was higher in the initial weeks PSO and then gradually decreased. Our results are consistent with germinal center formation required for antibody maturation. Germinal centers are rapidly formed upon encountering a new antigen and last for several weeks (up to 100 days) [26,27,28,29]. Our results indicate that the decline of anti-RBD levels observed starting six weeks post-symptom onset is somehow compensated by the presence of antibodies presenting higher avidity.

### 3.4. Evolution of RBD Avidity in SARS-CoV-2 Vaccinated Individuals

To evaluate whether mRNA vaccination resulted in a similar improvement of the avidity of vaccine-elicited anti-RBD Abs, we analyzed plasma samples from a cohort of 26 SARS-CoV-2 naïve donors (11 males and 15 females, median age of the donors: 50 years [range: 21–62 years]) after an initial dose of the Pfizer BioNTech mRNA vaccine (Figure 3A). These donors received their second dose after 16 weeks (extended interval vaccination regimen), allowing us to measure the avidity of vaccine-elicited anti-RBD IgG 3 weeks (median [range]: 21 days [16–28 days]) and 12 weeks (median [range]: 83 days [67–92 days]) after the first dose [12]. Three weeks post-vaccination, the anti-RBD avidity was low, but this level significantly increased 12 weeks post-vaccination (Figure 3B). Comparisons of anti-RBD IgG levels 12 weeks PSO or post-vaccination revealed a significantly higher amount of anti-RBD IgG following natural infection (Figure 3C). Nevertheless, when we measured the RBD avidity index in convalescent donors 12 weeks PSO (20 donors, median [range]: 85 days [66–99 days]) and in naïve donors 12 weeks post first dose of mRNA vaccine, we did not observe a significant difference in avidity index (Figure 3D), showing that affinity maturation is not dependent on specific Ab concentration.

### 3.5. RBD Avidity Correlates with B Cell Class Switch

Antibodies produced primarily shorter after SARS-CoV-2 infection are IgM with low affinity for the antigen. Somatic hypermutation in germinal centers accompanied the B cell class switch results in higher affinity IgG. We previously described how the frequency of RBD-specific IgM+ B cells decreased significantly PSO and conversely the frequency of RBD-specific IgG+ B cells increased between 6 and 21 weeks PSO and remained stable between 21 and 31 weeks [8]. Interestingly, we observed a negative correlation between the level of anti-RBD IgM+ B cell and the RBD avidity and a positive correlation between anti-RBD IgG+ B cell and the RBD avidity (Figure 4). These results reflect the class switch of B cells, associated with affinity maturation of individual antibodies leading to improved avidity measured in plasma samples.

## 4. Discussion

Humoral responses after the SARS-CoV-2 infection or vaccination are essential to limit and prevent infection. Although the level of Abs produced is an important parameter, the avidity of these antibodies also plays a crucial role, as it is associated with improved neutralization and potentially other antiviral functions of antibodies, such as antibody-dependent cellular cytotoxicity (ADCC) and other Fc-effector functions by facilitating interaction with its cognate epitope. The increase in the avidity is the results of somatic hypermutations that occur over time in the germinal centers [30,31]. In this instance, we longitudinally analyzed the RBD avidity of Abs induced after SARS-CoV-2 infection. Although the level of IgG gradually decreased over time after infection, we observed that their overall affinity (avidity) significantly increased, a phenomenon that appear to stabilize by week 32, in agreement with the time associated with the natural contraction of germinal centers [28,29]. Interestingly, we observed that the increase in avidity observed after natural SARS-CoV-2 infection was similar to the one detected after the mRNA vaccination.

In our study, we measured the avidity of the IgG against the RBD, which is involved in viral transmission and neutralization. However, while neutralization is an important component of the humoral response, several studies suggest that other functions of Abs may play an important role in humoral responses to SARS-CoV-2, notably antibody-dependent cellular cytotoxicity (ADCC) and antibody-dependent cellular phagocytosis (ADCP) which may be mediated by antibodies recognizing other domains than the RBD [32,33,34]. Thus, it would be interesting to see whether avidity against other domains of the Spike also increases over time.

Donors included in our cohort had been infected during the first wave of COVID-19 (between March and May 2020) before the emergence of variants of concern (VOCs). It would be interesting to see whether avidity evolves in the same way in donors infected with different VOCs. It is also likely that vaccination, with vaccines developed with the original strain of Wuhan, allows an enhanced breadth of avidity in convalescent donors infected with a VOC compared to donors infected by the original strain, as well as in donors vaccinated and then infected with a variant.

Since vaccines against SARS-CoV-2 have become available, countries have adopted different vaccine strategies, including type of vaccine, dose interval, dose administration, and consideration of the individual’s pre-vaccination status (SARS-CoV-2 naïve or previously infected). Our data suggest that an extended interval (16 weeks) between the first and second dose of the vaccine would allow time for superior affinity maturation (and improved germinal center formation) before the second contact with the antigen (second dose of vaccine), as recently suggested [12,35]. It would be interesting to observe how the avidity of plasma Abs measured after the second dose of the vaccine evolves compared to the short 3- or 4-week interval. These data would help refine optimal vaccination protocols. Accordingly, it has been demonstrated that individuals vaccinated with an extended interval have a better IgG avidity than individuals vaccinated with the recommended interval [12]. One clear advantage of the RBD avidity index assay, is that it is more straightforward and employs high throughput techniques, compared to other assays used to evaluate qualitative evolution of antibody responses. Therefore, we believe it exhibits strong potential as an immuno-monitoring tool.

Following the emergence of the Omicron variant, many countries have decided to administer a third dose of the vaccine, and some are in the process of approving a fourth dose. Our results suggest that monitoring antibody maturation and avidity in these different vaccine regimens is important to inform future vaccination campaigns.

## Figures and Tables

**Figure 1 viruses-14-00532-f001:**
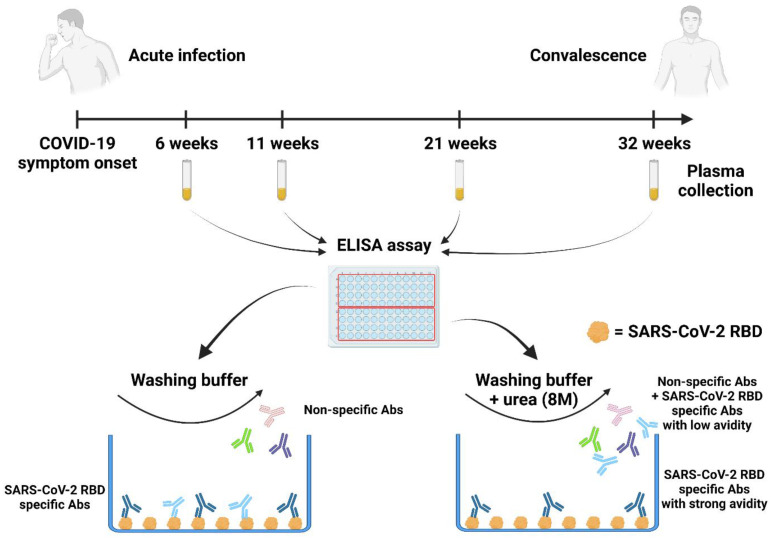
**RBD avidity assay**. Plasma samples were collected from a cohort of 29 COVID-19 recovered donors at 6, 11, 21, and 32 weeks PSO. 96-well ELISA plates were coated with SARS-CoV-2 S RBD proteins and incubated with plasma samples. Then, wells were washed with an ELISA buffer or an ELISA buffer supplemented with urea (8M). Created with BioRender.com.

**Figure 2 viruses-14-00532-f002:**
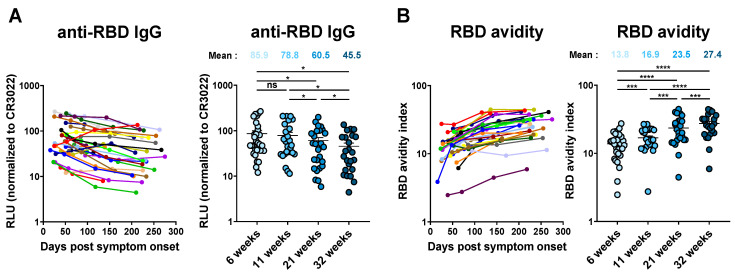
**RBD avidity of specific IgG increases over time in SARS-CoV-2 convalescent individuals.** (**A**) Level of RBD-specific IgG was measured using an indirect ELISA. Anti-RBD Ab binding was detected using HRP-conjugated anti-human IgG. Relative light unit (RLU) values obtained were normalized to the signal obtained with the anti-RBD CR3022 mAb present in each plate. (**B**) An indirect ELISA was performed to calculate the RBD avidity index. (**A**,**B**) In the left panels, each curve (shown in a different color) represents the values obtained with the plasma of one donor at every time point tested in triplicate, and in the right panels, plasma samples were grouped in different time points (6, 11, 21, and 32 weeks PSO). (**A**,**B**) significance was tested using a repeated measures one-way ANOVA with a Holm–Sidak post-test. Error bars indicate means ± SEM. (* *p* < 0.05; *** *p* < 0.001; **** *p* < 0.0001; ns, Statistical non-significant).

**Figure 3 viruses-14-00532-f003:**
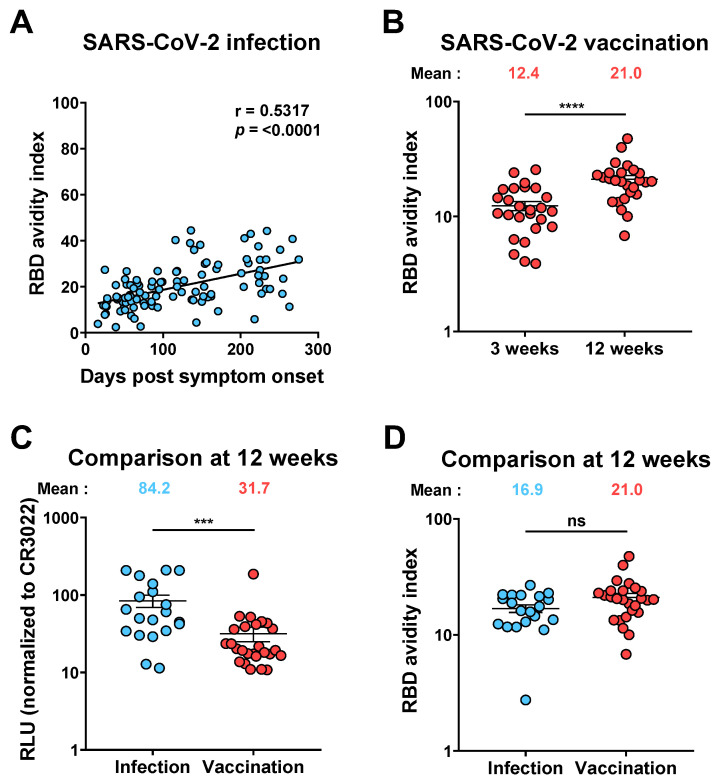
**RBD avidity increases with time after SARS-CoV-2 infection and vaccination.** (**A**) A positive correlation was found between time and avidity in convalescent donors after SARS-CoV-2 infection. (**B**) RBD avidity index measured in naïve vaccinated donors 3 and 12 weeks after the first dose of SARS-CoV-2 mRNA vaccine. Comparison of the anti-RBD IgG levels (**C**) and the RBD avidity index (**D**) at 12 weeks PSO or post vaccination. Convalescent and vaccinated donors are represented by blue and red points, respectively. Statistical significance was tested using Spearman correlation test (**A**), Wilcoxon paired t-test (**B**) and Mann–Whitney unpaired t-test (**C**,**D**). Error bars indicate means ± SEM. (*** *p* < 0.001; **** *p* < 0.0001; ns, non-significant).

**Figure 4 viruses-14-00532-f004:**
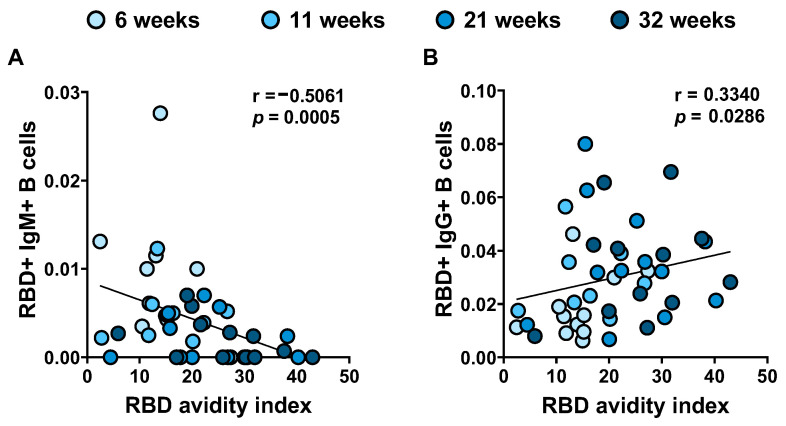
**Correlations between RBD-specific B cells and RBD avidity index over time post-symptom onset**. For 12 donors, characterization of RBD-specific B cells was monitored on longitudinal PBMC samples obtained from COVID-19+ convalescent individuals [8]. RBD avidity inversely correlates with RBD-specific IgM+ B cells (**A**) and correlates with RBD-specific IgG+ B cells (**B**). Statistical significance was tested using Spearman correlation tests.

**Table 1 viruses-14-00532-t001:** Longitudinal COVID-19 convalescent cohort.

Group	*n*	Days PSO (Median; Day Range)	Age (Median; Age Range)	Male (*n*)	Female (*n*)
**6 weeks**	29	45 (16–95)	48 (21–65)	15	14
**11 weeks**	25	77 (48–127)	48 (21–65)	14	11
**21 weeks**	26	146 (116–171)	49 (21–65)	14	12
**32 weeks**	24	226 (201–275)	51 (21–65)	15	9

## Data Availability

The data is contained within the article and Appendix A.

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
