# Peer review of "Evolution of Anti-RBD IgG Avidity following SARS-CoV-2 Infection"

_viruses, 2022, doi:10.3390/v14030532_

Round 1
Reviewer 1 Report
Congratulation is a well written and interesting article.
Have you diagnosed different variants of COVID-19?
Could variants have a different immunological behaviour?
Author Response
Please see attached rebuttal.

Reviewer 2 Report
It is a very interesting article to read. It provided timely and valuable data to the COVID-19 management. Authors developed the simple and straightforward method to help monitor the antibody response. Few questions need authors to clarify:
- Have the author verified the quality and purity of SARS-CoV-2 S RBD protein?
- In the results section, 3.1, authors listed the process and criteria for the sample collection. I think it is better to put these in the Material and Methods section.
- How many replicates of ELISA analysis did the authors do?
- In Figure 2, in the left panel of B, why the y-axis presented as RBD avidity index, not the RLU normalized results? Can you please clarity why?
- In the discussion, authors mentioned that "Our data suggest that an extended interval (16 weeks) between the first and...with the antigen". Please clarify how did you give 16 weeks conclusion? In your results you only compared at 3 and 12 weeks PSO and post vaccination.
Author Response
Please see attached rebuttal.

Reviewer 3 Report
This manuscript describes the progressive increase in anti-RBD IgG avidity after SARS-CoV-2 infection or vaccination. This is a well written manuscript with robust statistical analysis to support the case. I have two minor concerns which authors should address to improve the merit of the manuscript.
- Authors should provide gating strategy for the flow cytometry experiment.
- The details of cohort should be provided for comprehensive analysis of data. For example, if there was any comorbidities in the cohort.
Author Response
Please see attached rebuttal.
